# Age-Related Variation of Pulpal Oxygen Saturation in Healthy Primary and Permanent Teeth in Children: A Clinical Study

**DOI:** 10.3390/jcm12010170

**Published:** 2022-12-26

**Authors:** Andreea Igna, Darian Rusu, Emilia Ogodescu, Ștefania Dinu, Marius Boariu, Adrian Voicu, Ștefan-Ioan Stratul

**Affiliations:** 1Department of Pediatric Dentistry, Pediatric Dentistry Research Center, Faculty of Dental Medicine, “Victor Babes” University of Medicine and Pharmacy, 300041 Timisoara, Romania; 2Department of Periodontology, Faculty of Dental Medicine, Anton Sculean Research Center for Periodontal and Peri-Implant Diseases, “Victor Babes” University of Medicine and Pharmacy, 300041 Timisoara, Romania; 3Department of Endodontics, Faculty of Dental Medicine, TADERP Research Center, “Victor Babes” University of Medicine and Pharmacy, 300041 Timisoara, Romania; 4Department of Functional Sciences, Faculty of Medicine, “Victor Babes” University of Medicine and Pharmacy, 300041 Timisoara, Romania

**Keywords:** pulse oximetry, oxygen saturation, primary teeth, permanent teeth, root development, tooth type

## Abstract

(1) Background: Pulse oximetry (PO) is an effective method of dental pulp status monitorization but still lacks practical implementation in dentistry, as well as clear reference values for different tooth types. The study’s aim was to investigate the age-related variation of blood oxygen saturation (SpO_2_) from the dental pulp during different stages of tooth development in all types of primary and permanent teeth of children. (2) Methods: The pulps of 600 healthy primary and permanent teeth (incisors, canines, premolars, and molars) of patients aged 2–15 years were tested with an adapted PO device, and the results were statistically analyzed; (3) Results: Statistically significant differences (*p* < 0.05) were found between open-apex and closed-apex teeth and between the canines and other tooth types in both primary and permanent dentitions. (4) Conclusions: Values of SpO_2_ tended to decrease with age progression in both primary and permanent dentitions. Enamel and dentine thickness and their optical properties and the shape and volume of coronal pulp, which differed among tooth types, seemed to have some influence on the reading as well. The study indicates that factors such as the root development and the tooth type must be taken into account when establishing reference SpO_2_ values for the dental pulp.

## 1. Introduction

Pulse oximetry (PO) is an effective and objective oxygen saturation monitoring technique, broadly used in medicine for recording blood oxygen saturation levels, using finger, toe, foot, and ear probes [1,2,3]. A PO probe consists of two light-emitting diodes (LEDs), which transmit red light energy (660 nm) and infrared light energy (940 nm), and a photodetector diode connected to a signal-processing unit [4]. The pulse oximeter correlates the information with the known absorption curves for oxygenated and deoxygenated hemoglobin, which absorb different amounts of red and infrared lights, to determine oxygen saturation (SpO_2_) levels [5]. PO is also used in endodontics for differential diagnosis between vital and necrotic dental pulps [6]. A dedicated dental pulse oximeter is not commercially available for clinical use yet [2], but multiple custom-made devices have been investigated for this purpose [7,8,9]. For an efficient dental use, the PO probe shape must ensure that the transmitter and the detector are parallel and well-fixed onto the tooth. Currently, the use of PO is not possible for very small primary teeth (such as the inferior incisors) and erupting teeth because of the large probe size. However, due to its non-invasive, atraumatic nature and relatively simple technique, PO is a suitable diagnostic method for pediatric use [5,10], overcoming some of the drawbacks of sensitivity pulp tests (thermal or electrical), such as pain triggering (which has a negative impact on child cooperation), or unreliability due to the inability of little children to accurately report sensation [11,12]. In children, dental trauma and deep carious lesions are frequently treated by vital pulp therapy procedures, which aim to preserve the integrity and health of the teeth and their supporting tissues while maintaining the vitality of the pulp [13]. In this regard, the pulse oximeter could be a useful tool for post-operatory monitorization of the pulp status to assess the success of vital pulp therapy procedures. Recently, an experimental study confirmed the diagnostic value of PO for pulp status evaluation in primary teeth, with histologic means, following pulp-capping in a canine model [14]. Other studies reported successful use of PO for monitorization of the pulp status during orthodontic treatment [15], following dental bleaching techniques [16], in patients with systemic conditions such as sickle cell anemia [17] or patients undergoing head and neck radiotherapy [18].

Reference SpO_2_ values for different tooth types are not yet established in the literature, due to the heterogeneity found in most studies, regarding patient cohorts and dental samples [2,3]. Furthermore, most studies were carried out in adult patients [19] and anterior teeth due to PO sensor size limitations [20,21,22]. For primary teeth, the data are scarce and conflicting [23,24,25,26]. To our knowledge, there is a single study which investigated the SpO_2_ of the pulpal blood flow (PBF) of primary teeth in relation to root physiological resorption; however, this study was limited to incisors [26].

Throughout their lifetime, from development to exfoliation, primary teeth undergo functional and structural changes. A study of the physiological root resorption in primary teeth has revealed that although some changes do occur in the pulpal status of primary teeth within the resorption stage, these are not as profound as previously thought, therefore speculating that teeth could retain the potential for sensation, healing, and repair until advanced stages of root resorption [27]. Regarding the blood supply of primary teeth, the reported data are heterogenous. Karayilmaz et al. reported increased PBF in primary teeth with root resorption, attributed to the progressive apical enlargement [26], while Komatsu et al. reported decreased PBF in primary teeth with age progression, related to the morphological changes in the blood vessels in the pulp [28]. 

In immature permanent teeth, the incomplete development of Raschkow’s neural plexus until the late stages of root development has implications for the diagnosis of pulp vitality [29], which are sensitivity tests being unreliable during apexogenesis. It is now widely acknowledged by specialists that immature permanent teeth, as well as traumatized teeth, give more accurate and reliable responses to vitality testing (using PO or laser Doppler flowmetry (LDF)), rather than to sensibility testing (thermal or electrical) [30,31,32]. Multiple systematic reviews on pulp tests have already confirmed the superiority of vitality pulp tests as diagnostic means, highlighting the need for further studies to investigate them in different clinical circumstances [2,19,31,33]. 

The aim of our study was to investigate the age-related variation of blood oxygen saturation in the dental pulp during different stages of tooth development in all types of primary and permanent teeth (incisors, canines, premolars, and molars) of 2−15-year-old children.

## 2. Materials and Methods

The present study was carried out on 200 patients of the Pediatric Dentistry Clinic from the University of Medicine and Pharmacy “Victor Babes” from Timisoara, Romania, with the approval of the university’s ethics committee (approval nr. 51/2020 of UMFVBT). Written informed consent was given by the legal tutors of all children involved in the study. The patients were 115 males and 85 females, aged between 2 and 15 years (Figure 1). 

Oxygen saturation readings of the dental pulp and the nasal wing were performed. The SpO_2_ of the dental pulp was measured in 650 teeth, divided into five groups: group I (G1)—primary teeth without physiological resorption (closed apex): 50 incisors, 50 canines, and 100 molars (50 first primary molars and 50 s primary molars); group II (G2)—primary teeth with physiological resorption (open apex): 50 incisors, 50 canines, and 100 molars (50 first primary molars and 50 s primary molars); group III (G3)—immature permanent teeth with an open apex: 50 incisors, 50 canines, 50 premolars, and 50 first permanent molars; group IV (G4)—young permanent teeth with a closed apex: 50 incisors, 50 canines, 50 premolars, and 50 first permanent molars; group V (G5)—50 endodontically treated teeth (control). The degree of root development/ resorption was determined radiographically on ortopantomographs (OPGs) (Figure 2).

The inclusion criteria for G1−G4 were as follows: healthy children, intact healthy teeth (with no clinical or radiological signs of pathology and without a history of trauma), and teeth with an adequate size to allow an optimal placement of a PO sensor; the exclusion criteria were as follows: children with systemic diseases which could influence oxygen saturation levels, teeth with clinical or radiological signs of pathology, teeth previously affected by dental trauma, teeth with coronal restorations, and teeth with advanced physiological root resorption (less than 1/3 root present). The inclusion criterion for G5 was as follows: endodontically treated teeth. The SpO_2_ readings were performed using a portable pulse-oximeter (SOMO PO100; SOMO International CO., Ltd., Hong Kong, China) with a compatible pediatric nasal alar sensor (Nasal Alar Fast SpO₂ Sensor; Koninklijke Philips N.V., Amsterdam, the Netherlands), as shown in Figure 2a. 

Two investigators (A.I. and M.B.) and a supervisor (E.O.) were involved in the process of patient/tooth selection (according to clinical and radiographical criteria) and pulp testing. The reading time/tooth was between 30 s and 2 min, depending on the child’s cooperation; a mean value was recorded for each measurement. The teeth were isolated with liquid rubber dam (SDI Gingival Barrier, SDI Ltd., Chicago, IL, USA) placed circularly on the adjacent marginal gingiva (Figure 3b,c). The probe was hand-stabilized by the operator during the measurement, and the operating light from the dental unit was switched off. 

The SpO_2_ readings of the dental pulp divided into the five groups were compared to each other and also with SpO_2_ readings performed on the patients’ nasal wings. Data were statistically analyzed using parametric and non-parametric tests. A *p* value of <0.05 was considered statistically significant. In some situations, we also considered the effect size ε² as a qualitative assessment of the effects (values of >0.5 can be considered as a medium-to-large effect). The data were processed with the statistical software R Core Team Version 4.1 (2021), R ggplot2 package Version 3.4.0, and JAMOVI Version 2.3 (2022). 

## 3. Results

The results of the statistical analysis of the recorded SpO_2_ values revealed several significant differences between teeth in different developmental stages. In primary teeth, values of vital pulp ranged between 72% (min.) and 98% (max.), with a mean value of 90% for teeth with a closed apex (in the stability stage) and a mean value of 89% for teeth with an open apex (in the resorption stage); in permanent teeth, values of the vital pulp ranged between 77% (min.) and 94% (max.), with a mean value of 91.6% for teeth with an open apex (immature) and a mean value of 86.4% for teeth with a closed apex (mature), as shown in Figure 4 and Table 1. The oxygen saturation readings obtained from the teeth were significantly lower when compared to from the nasal wing (mean: 98.8%; min.: 96%; max.: 100%). Endodontically treated teeth in the G5 control group (non-vital) recorded no oxygen saturation (value: 0%). 

Four rounds of comparisons were performed on the G1−G4 groups as follows: closed apex vs. open apex (G1 vs. G2 and G3 vs. G4); primary vs. permanent teeth (G2 vs. G3 and G1 vs. G4); tooth types (incisors, canines, premolars, and molars) against each other within each of the four groups (G1−G4); upper vs. lower teeth of each type within each of the four groups (G1−G4). 

Comparisons between primary teeth and permanent teeth according to their apex type—open/closed (G1 vs. G2 and G3 vs G4) were carried out using the Mann−Whitney U non-parametric test. Statistically significant differences between SpO_2_ according to the apex type were recorded in both primary teeth—G1 vs. G2 (*p* = 0.004) and permanent teeth —G3 vs. G4 (*p* < 0.001), but the effect size was much higher for permanent teeth (ε² = 0.640) than for primary teeth (ε² = 0.167). Values of SpO_2_ tended to decrease with age progression in both primary and permanent teeth (Figure 5).

Comparisons between open-apex primary and permanent teeth (G2 vs. G3) and closed-apex primary and permanent teeth (G1 vs. G4) were performed using the Mann−Whitney U non-parametric test. Significant differences were recorded in both cases (*p* < 0.001; Figure 6), with an effect size slightly more significant for the closed-apex groups (ε² = 0.470) than for the open-apex groups (ε² = 0.346). 

Comparisons between tooth types (incisors, canines, premolars, and molars) within each group (G1, G2, G3, and G4) were carried out using the Kruskal−Wallis non-parametric test and Dwass−Steel−Critchlow−Fligner pairwise comparisons. In primary teeth, the tests revealed statistically significant differences (*p* < 0.001) only between the canines and the other tooth types (incisors and molars; Table 2), but with small effect sizes of ε² = 0.152 for the teeth without physiological resorption (closed apex) and ε² = 0.093 for the teeth with physiological resorption (open apex). No statistically significant differences were registered when comparing first primary molars to second primary molars. 

In permanent teeth with an open apex, the most statistically significant differences (*p* < 0.005) were recorded when comparing canines to other tooth types (incisors, molars, and premolars) and premolars to other tooth types (incisors, canines, and molars), while in teeth with a closed apex, significant differences were registered only in the canine comparisons (*p* < 0.001), as shown in Table 3. The effect sizes of ε² = 0.203 (open apex) and ε² = 0.135 (closed apex), however, were relatively small.

Comparisons between superior and inferior teeth were performed for each tooth type (except the incisors), within each group (G1, G2, G3, and G4), using the Mann−Whitney U non-parametric test. The results obtained in the primary teeth groups (G1 and G2; Table 4) revealed significant differences only in the canine comparisons, within the open-apex group G2 (*p*=0.010, ε² = 0.433; Figure 7). No statistically significant results were obtained in the permanent teeth groups (G3 and G4; Table 5). Descriptive statistical data are found in Table 6. 

## 4. Discussion

This study, carried out on children’s primary and permanent teeth, aimed to assess the variation of the dental pulp’s blood oxygen saturation of different tooth types during different stages of tooth development. Regarding the influence of age on pulp vitality results using pulse oximetry, our findings indicate that SpO_2_ tended to decrease with age progression in both primary and permanent dentitions. This decrease was more significant in primary incisors and canines. While in primary teeth the SpO_2_ decreased during an apex opening caused by physiological resorption, in permanent teeth SpO_2_ decreased with an apex closure. Our results are in accordance to those obtained by Komatsu et al. in human primary incisors, who related the decrease of PBF to the morphological changes in the blood vessels in the pulp [28], but in contrast with those obtained by Karayilmaz et al., who reported an increase of the pulpal blood flow in primary teeth with age, attributed to the progressive apical enlargement caused by physiological root resorption [26]. The decrease in SpO_2_ with age progression and concomitant apical enlargement in primary teeth may suggest that the reading could be directly correlated to the pulpal status and be not affected by the communication of the pulp with the periapical tissue. 

In case of permanent teeth, it is known that in younger patients the levels of saturation are higher than in those of greater age [34,35]. The mean SpO_2_ levels we found in anterior permanent teeth in children aged 7−15 years (incisors: 92.3–87.1%; canines: 88–83%) are consistent with the results of other studies for approximately the same age group (incisors: 87.7% [24], 87.1% [36], and 84.3% [34]; canines: 83.4% [24]). For posterior teeth, the available data are scarcer [25,35,37]. A study investigating SpO_2_ of maxillary premolars in different age groups (patients between 20 and 44 years) found that the older the patients, the lower the SpO_2_ is, even in the absence of pulp tissue injury, with mean values ranging between 89.7% (in the 20−24-year-old group) and 80.0% (in the 40−44-year-old group) [35]. In addition to this study, our results (in patients between 7 and 15 years) show a mean SpO_2_ of 93.3% in immature (open apex) premolars, which dropped to 89.3% once the apex was closed. We found the same tendency of the SpO_2_ decrease with an apex closure in all tooth types (incisors, canines, premolars, and molars), which is in accordance to what another recent study showed on maxillary incisors with open and closed apices [5]. This observation is important, when reference ranges of SpO_2_ levels are to be established for both healthy and affected dental pulps (inflamed and necrotic). As SpO_2_ values of healthy pulp largely differ in young individuals compared to in old individuals [3,19], if we consider a value of 70% in a tooth of a 60-year-old patient it is within the physiological limits of a healthy pulp, while the same value registered in an immature permanent tooth could be indicative of a pulp disease. These limits are narrower in age groups that are closer to each other, like in our study. We believe that the significant differences we obtained between open-apex and closed-apex teeth will become truly clinically relevant in the future, when more data on SpO_2_ levels in different pulp diseases, according to each age group, are available. The evidence [2,19,20] suggests that there might be superimpositions of values that can be considered indicative for both healthy pulp in one age group and diseased pulp in another age group. In-depth knowledge on age-related SpO_2_ variation is therefore essential, in order to avoid misinterpretations. No correlation between the dental and alar SpO_2_ was observed in our study. The saturation measured at the nose wing in all patients was higher than in the tested teeth, a fact that can be explained by the different thickness of diffraction media, which in case of the teeth are enamel and dentin and in case of the nasal wing are skin and cartilage. These findings are in accordance with previous studies who compared dental and index finger SpO_2_ [4,24,37,38]. 

Due to the lack of a standardized method of SpO_2_ measurement for the dental pulp, there are a few factors known to influence the reading, such as probe design, ambient light and tissue thickness, which need to be taken into consideration [2]. For our study, we selected a pediatric nasal probe, with parallel sensors soldered together by a flexible clamp, which facilitated adaptation to the tooth. The probe was soft and easily accepted by patients of young age. Most previous studies were carried out predominantly on anterior teeth [3], due to limitations imposed by the size of the sensor, usually custom-modified from a finger probe [38,39]. The nasal probe used in our study had a small size, which made it appropriate for dental use, even in the posterior area of the oral cavity, ensuring a satisfactory tooth-fit. The sensor produced a strong, consistent signal, even in patients with poor perfusion, being recommended for this feature by other authors [40]. Nevertheless, the width of the sensor was larger than the mesio-distal width of the primary inferior incisors, a fact that made measurements in these teeth unreliable. Thus, primary inferior incisors were excluded from our investigation. This represents a limitation of our study. A drawback of the technique used was the need for probe stabilization onto the tooth, which was performed by the hand of the operator, especially needed in lateral areas of the lower arch, to impede the tongue movements. However, the movement of the probe onto the tooth surface is a common artifact, and one solution to lower it is to record the mean signal over time [41], which we did by maintaining the probe in position for as long as possible (between 30 s and 2 min), depending on the child’s cooperation. Ambient light is considered by some authors to influence the SaO_2_ reading to some degree [42], but there are also authors whose results show that PO can be used in practice without the need for gingival tissue isolation [43]. In our study, we used a minimal isolation for the marginal gingiva—liquid rubber dam. On the other hand, tissue thickness (in case of teeth enamel and dentin) interferes with SaO_2_, regardless of the presence or absence of ambient light, the lowest SaO_2_% levels being registered in the thickest tooth samples [42,44]. The same effect was seen with LDF pulp testing, in which signal strength is lower when the penetrated tissues are thicker [45]. Furthermore, dentine’s optical properties, such as color and translucency, were found to differ among tooth types [46]. These factors, along with the differences in shape and volume of the pulp chambers of different tooth types, might provide an explanation for the SpO_2_ differences we found between canines and the other tooth types in our study. 

Studies on the use of pulse oximetry in dentistry are highly heterogenic because of different methodologies used, control of confounding factors, sample size, tooth type, and patient age [3]. Most previous studies excluded children due to difficult behavior management during the procedure [19]. Nevertheless, our study confirmed the fact that the patient’s age is an important factor affecting the results, and a major indication of vitality diagnostic pulp tests is particularly for children with traumatized immature teeth, which, according to other authors, exhibit a high potential for regeneration [30]. 

The results we obtained might contribute to the establishment of SpO_2_ reference parameters for all types of primary teeth, with and without physiological resorption, and for immature and mature permanent teeth in the 7–15-year-old group. Reference SpO_2_ parameters for the healthy pulp are extremely necessary for the diagnosis of various pulp conditions, as well as for the clinical decision-making process in case of traumatized teeth and teeth treated by vital pulp therapy or regenerative endodontic therapy (e.g., re-vascularization). Unlike previous studies, our research focused on the variation of SpO_2_ in the context of root development, which is a key factor for all pediatric dental procedures. Within the limitations of the present study (the lack of a customized probe for dental use, large sensor size, and hand for the stabilization of the probe), we can state that the root development and the tooth type have a significant influence on SpO_2_ values for the dental pulp. Standardized clinical protocols and improvements in pulse oximeter technology are highly needed by researchers to accommodate diagnostic needs in oral environments and to allow designing of high-quality studies to establish reliable reference pulpal SpO_2_ values to be used in clinical practice. 

## 5. Conclusions

Values of SpO_2_ tended to decrease with age progression in both primary and permanent dentitions. The decrease seemed to be more significant in case of the permanent teeth. The thickness and the optical properties of hard dental tissues (enamel and dentin), as well as the shape and volume of the coronal pulp, which differed among tooth types, seemed to have some influence on the reading as well. The present study indicates that factors such as the root development and the tooth type must be taken into account when establishing reference SpO_2_ values.

## Figures and Tables

**Figure 1 jcm-12-00170-f001:**
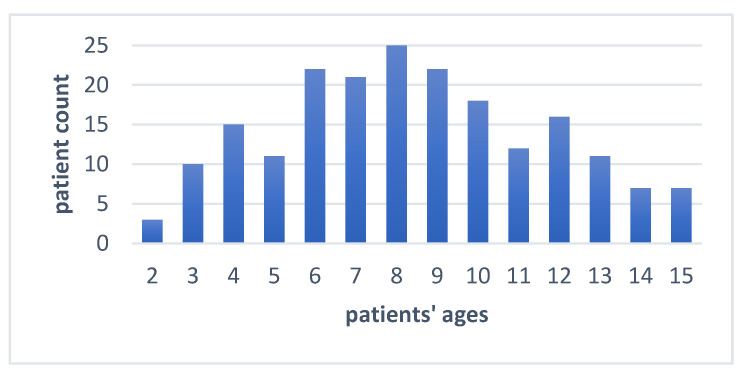
Graphic representation of the patients’ distribution according to their age.

**Figure 2 jcm-12-00170-f002:**
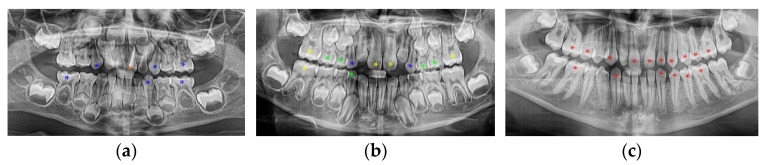
OPGs of children aged 3 years old (**a**), 8 years old (**b**), and 15 years old (**c**). The teeth considered suitable for SpO_2_ measurements were marked with color-coded asterisks (*) as follows: G1 is indicated in blue, G2 is indicated in green, G3 is indicated in yellow, G4 is indicated in red, and G5 is indicated in orange.

**Figure 3 jcm-12-00170-f003:**
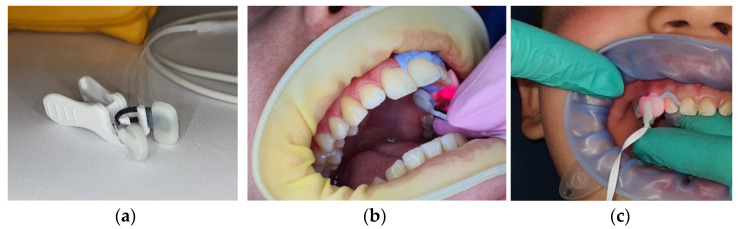
(**a**)The small alar sensor together with a fixation clamp; (**b**) SpO_2_% determination in a permanent superior canine; (**c**) SpO2% determination in a primary superior canine.

**Figure 4 jcm-12-00170-f004:**
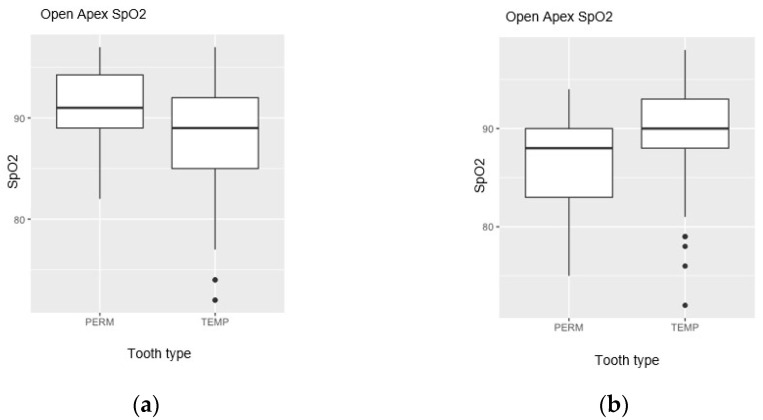
Box plots showing the distributions of SpO_2_ values of permanent (PERM) and primary (TEMP) teeth with an open apex (**a**) and with a closed apex (**b**) around the median value (box) and outliers (dots).

**Figure 5 jcm-12-00170-f005:**
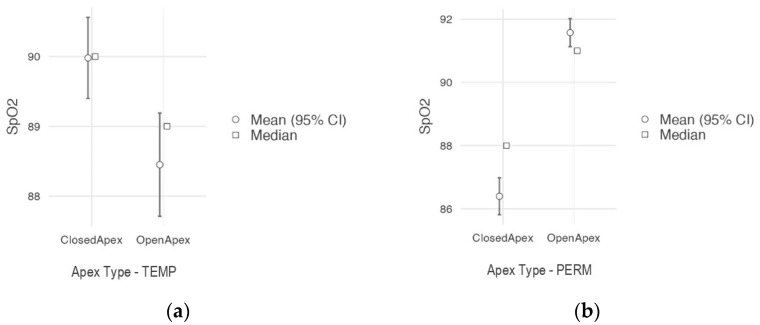
Graphic representations of SpO_2_ measured in teeth with an open vs. a closed apex: (**a**) primary teeth; (**b**) permanent teeth.

**Figure 6 jcm-12-00170-f006:**
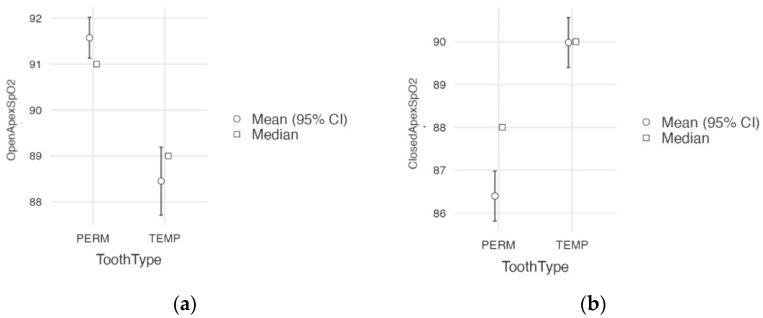
Graphic representations of SpO_2_ measured in primary vs. permanent teeth with an open apex (**a**) and a closed apex (**b**).

**Figure 7 jcm-12-00170-f007:**
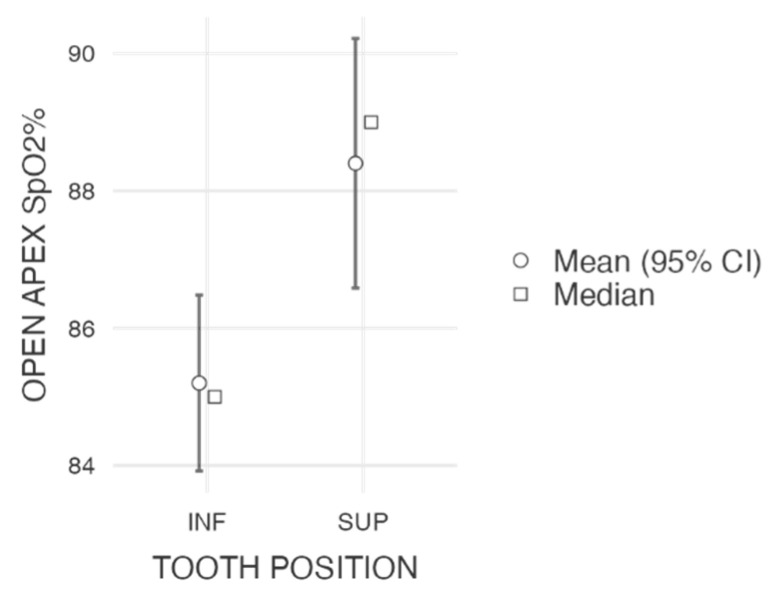
Graphic representation of SpO_2_ measured in primary superior (SUP) vs. inferior (INF) canines with an open apex.

**Table 1 jcm-12-00170-t001:** SpO_2_ values of primary and permanent teeth by tooth type.

Apex Type	Tooth Type ^1^		N	Mean SpO_2_ (%)	Median SpO_2_ (%)	SD	Min. SpO_2_ (%)	Max. SpO_2_ (%)
Open apex	TEMP	I	50	86.6	87.0	7.5	72	97
		C	50	86.8	87.0	4.3	80	97
		M1	50	88.8	88.5	3.8	82	97
		M2	50	91.6	91.0	4.2	72	97
	PERM	I	50	92.3	92.0	2.5	88	97
		C	50	88.0	87.0	1.5	87	91
		PM	50	93.3	94.0	2.7	89	97
		M	50	92.7	93.0	2.9	72	97
Closed apex	TEMP	I	50	91.8	91.0	1.7	89	95
		C	50	87.7	88.0	4.1	78	97
		M1	50	89.2	90.0	4.8	76	98
		M2	50	91.2	91.0	4.2	72	97
	PERM	I	50	87.1	88.0	3.3	78	93
		C	50	83.0	82.0	4.0	77	90
		PM	50	89.3	90.0	2.7	81	93
		M	50	86.1	87.0	4.2	78	94

^1^ TEMP, primary; PERM, permanent; I, incisors; C, canines; M1, first primary molars; M2, second primary molars; PM, premolars; M, first permanent molars.

**Table 2 jcm-12-00170-t002:** Pairwise comparisons between primary teeth by tooth type.

Apex Type	Tooth type ^1^–Pairwise Comparisons	*p*
Open	I–C	0.995
	I–M	0.092
	C–M	**<0.001**
	M1–M2	0.986
Closed	I–C	**<0.001**
	I–M	0.121
	C–M	**<0.001**
	M1–M2	0.516

^1^ I, incisors; C, canines; M, molars (both first and second); M1, first primary molars; M2, second primary molars. Bold *p* Values indicate significant differences

**Table 3 jcm-12-00170-t003:** Pairwise comparisons between permanent teeth by tooth type.

Apex Type	Tooth Type ^1^–Pairwise Comparisons	*p*
Open	I–C	**<0.001**
	I–PM	**0 .005**
	I–M	0.955
	C–PM	**<0.001**
	C–M	**<0.001**
	PM–M	**0.002**
Closed	I–C	**<0.001**
	I–PM	0.910
	I–M	0.895
	C–PM	**<0.001**
	C–M	**<0.001**
	PM–M	0.999

^1^ I, incisors; C, canines; PM, premolars; M, molars. Bold *p* Values indicate significant differences

**Table 4 jcm-12-00170-t004:** Pairwise comparisons between superior and inferior primary teeth by tooth type.

Apex Type	Tooth Type ^1^ and Position ^2^–Pairwise Comparisons	*p*
Open	C sup.–C inf.	**0.010**
	M1 sup.–M1 inf.	0.106
	M2 sup.–M2 inf.	0.984
Closed	C sup.–C inf.	0.155
	M1 sup.–M1 inf.	0.066
	M2 sup.–M2 inf.	0.096

^1^ C, canines; M1, first primary molars; M2, second primary molars; ^2^ sup., superior; inf., inferior. Bold *p* Value indicate significant differences

**Table 5 jcm-12-00170-t005:** Pairwise comparisons between superior and inferior permanent teeth by tooth type.

Apex Type	Tooth Type^1^ and Position ^2^–Pairwise Comparisons	*p*
Open	C sup.–C inf.	0.937
	PM sup.–PM inf.	0.550
	M sup.–M inf.	0.550
Closed	C sup.–C inf.	0.206
	PM sup.–PM inf.	0.698
	M sup.–M inf.	0.513

^1^ C, canines; PM, premolars; M, molars. ^2^ sup., superior; inf., inferior.

**Table 6 jcm-12-00170-t006:** SpO_2_ values of primary and permanent teeth by tooth type and position.

Apex Type	Tooth Type ^1^		Tooth Position ^2^	N	Mean SpO_2_%	Median SpO_2_%	SD	Min. SpO_2_%	Max. SpO_2_%
Open apex	TEMP	C	sup.	25	88.4	89.0	4.64	80	97
		C	inf.	25	85.2	85.0	3.27	80	92
		M	sup.	25	89.8	90.0	3.82	83	97
		M	inf.	25	90.5	90.0	3.77	82	97
	PERM	C	sup.	25	87.8	89.0	2.58	82	92
		C	inf.	25	88.0	87.0	1.54	87	91
		PM	sup.	25	93.0	94.0	2.88	89	97
		PM	inf.	25	93.5	94.0	2.47	89	97
		M	sup.	25	93.0	93.0	2.89	87	97
		M	inf.	25	92.4	91	2.93	87	97
Clpsed apex	TEMP	C	sup.	25	86.9	86.0	3.43	81	94
		C	inf.	25	88.6	88.0	4.62	78	97
		M	sup.	25	90.1	90.0	5.15	76	98
		M	inf.	25	90.3	90.0	4.03	72	96
	PERM	C	sup.	25	84.4	85.0	4.09	75	90
		C	inf.	25	83.0	82.0	4.05	77	90
		PM	sup.	25	89.2	89.0	2.66	81	93
		PM	inf.	25	89.3	90.0	2.78	81	93
		M	sup.	25	85.7	87.0	4.84	78	94
		M	inf.	25	86.6	87.0	3.44	80	92

^1^ C, canines; PM, premolars; M, molars. ^2^ sup., superior; inf., inferior.

## Data Availability

Not applicable.

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
