# Peer review of "Age-Related Variation of Pulpal Oxygen Saturation in Healthy Primary and Permanent Teeth in Children: A Clinical Study"

_jcm, 2022, doi:10.3390/jcm12010170_

Round 1
Reviewer 1 Report
Thank you for the opportunity to review the article “Age-related variation of pulpal oxygen saturation in healthy primary and immature permanent teeth. A clinical study”. The Authors of the current manuscript aim to evaluate the age-related variation of blood oxygen saturation from the dental pulp during various stages of tooth development in all types of primary and permanent teeth. This is a large scale-study that addresses a clinically important question.
My major comments/questions:
1. The title of the manuscript is not accurate “… in healthy primary and immature permanent teeth”, as the authors evaluate both immature permanent teeth with open apex and young permanent teeth with a closed apex. Please, clarify this.
2. Introduction section, line 47; “currently, the use of PO is not possible for very small primary teeth (like inferior incisors) … because of the large probe size.” How did you solve this problem since primary incisors were included in the study?
3. What type of radiographs were chosen to assess the degree of root development/resorption? How many investigators were involved in this process? I would recommend including a sample radiographic image for each group.
4. Results section, line 148; “this fact can be explained by the different thickness of the diffraction media … “. The interpretation of the results should be moved to the Discussion section.
5. The authors found that the blood oxygen saturation values of the dental pulp tend to decrease with age in both primary and permanent dentitions. Although this result is statistically significant, do you think that a reduction in blood oxygen saturation values of this magnitude is also clinically relevant? Please, discuss this.
Author Response
Dear Reviewer,
Thank you for taking the time to review our manuscript and for your kind observations. We greatly appreciate the suggestions made to improve our paper.
We have addressed your suggestions as follows:
- The title of the manuscript is not accurate “… in healthy primary and immaturepermanent teeth”, as the authors evaluate both immature permanent teeth with open apex and young permanent teeth with a closed apex. Please, clarify this.
We modified the title: “Age-related variation of pulpal oxygen saturation in healthy primary and permanent teeth in children. A clinical study”.
- Introduction section, line 47; “currently, the use of PO is not possible for very small primary teeth (like inferior incisors) … because of the large probe size.” How did you solve this problem since primary incisors were included in the study?
Unfortunately, we were not able to solve this problem, which remains a limitation of our study; although we selected the smallest probe we could find, still the width of the two sensors within the probe was larger than the primary inferior incisors, which is why we excluded them from the study. We added the following statements to the Discussion section to reflect this issue more clearly: ”Nevertheless, the width of the sensor is larger than the mesio-distal width of the primary inferior incisors, a fact that makes measurements in these teeth unreliable. Thus, primary inferior incisors were excluded from our investigation. This represents a limitation of our study.” (Line 329),”large sensor size” (Line 370).
- What type of radiographs were chosen to assess the degree of root development/resorption? How many investigators were involved in this process? I would recommend including a sample radiographic image for each group.
Thank you for the question. The following completions were added to the Material and Methods section: “The degree of root development/ resorption was determined radiographically, on ortopantomographs (OPGs) – Figure 2.” (Line 114) and “Two investigators (A.I., M.B.) and a supervisor (E.O.) were involved in the process of patient / tooth selection (according to the clinical and radiographical criteria) and pulp testing.” (Line 126).
We added a new figure -Figure 2, containing sample radiographic images that we used for patient / tooth selection, which include teeth from all 5 study groups.
- Results section, line 148; “this fact can be explained by the different thickness of the diffraction media … “. The interpretation of the results should be moved to the Discussion section.
We moved the phrase to the Discussion section (Line 315).
- The authors found that the blood oxygen saturation values of the dental pulp tend to decrease with age in both primary and permanent dentitions. Although this result is statistically significant, do you think that a reduction in blood oxygen saturation values of this magnitude is also clinically relevant? Please, discuss this.
We added the following fragment to the Discussion section: ”This observation is important when reference ranges of SpO2 levels are to be established for both healthy and affected dental pulps (inflamed and necrotic). As SpO2 values of healthy pulp largely differ in young compared to old [3,19], if we consider a value of 70%, in a tooth of a 60-year old patient it is within the physiological limits of a healthy pulp, while the same value registered in an immature permanent tooth could be indicative of a pulp disease. These limits are narrower in age groups that are closer to each other, like in our study. We believe that the significant differences we obtained between open-apex and closed-apex teeth will become truly clinically relevant in the future, when more data on SpO2 levels in different pulp diseases, corresponding to each age group, will be available. The evidence [2,20,38] suggests that there might be superimpositions of values that can be considered indicative for both healthy pulp in one age group, and diseased pulp in another age group. In-depth knowledge on age-related SpO2 variation is therefore essential, in order to avoid misinterpretations.” (Line 300).
We are grateful for your time and effort to review our paper and we hope we have successfully addressed all your queries!
Sincerely,
The authors
Reviewer 2 Report
Overall: In contrast to existing studies, the analysis focusing on tooth type and root development is commendable. I think this study has a contribution to the standardization of non-invasive vital tests in future pediatric dental medicine.
Point 1: Because you compare SpO2 between upper and lower teeth for Table 4 and 5, I think it will be better to show the separate low data of the upper and lower teeth.
Point 2(minor): More detailed explanation of the numbers, ex. 62, 63… in small rectangular boxes, may be needed in Legend in Figure 3.
Author Response

(The authors gave the same response as above.)
